# Tim29 is a novel subunit of the human TIM22 translocase and is involved in complex assembly and stability

**Yilin Kang[1,2], Michael James Baker[1,2], Michael Liem[3,4], Jade Louber[1,2], Matthew McKenzie[5], Ishara Atukorala[3,4], Ching-Seng Ang[2], Shivakumar Keerthikumar[3,4], Suresh Mathivanan[3,4], Diana Stojanovski[1,2]\***

[1]Department of Biochemistry and Molecular Biology, The University of Melbourne, Melbourne, Australia; [2]The Bio21 Molecular Science and Biotechnology Institute, The University of Melbourne, Melbourne, Australia; [3]Department of Biochemistry and Genetics, La Trobe University, Melbourne, Australia; [4]La Trobe Institute for Molecular Science, La Trobe University, Melbourne, Australia; [5]Centre for Genetic Diseases, Hudson Institute of Medical Research, Melbourne, Australia

**Abstract** The TIM22 complex mediates the import of hydrophobic carrier proteins into the mitochondrial inner membrane. While the TIM22 machinery has been well characterised in yeast, the human complex remains poorly characterised. Here, we identify Tim29 (C19orf52) as a novel, metazoan-specific subunit of the human TIM22 complex. The protein is integrated into the mitochondrial inner membrane with it's C-terminus exposed to the intermembrane space. Tim29 is required for the stability of the TIM22 complex and functions in the assembly of hTim22. Furthermore, Tim29 contacts the Translocase of the Outer Mitochondrial Membrane, TOM complex, enabling a mechanism for transport of hydrophobic carrier substrates across the aqueous intermembrane space. Identification of Tim29 highlights the significance of analysing mitochondrial import systems across phylogenetic boundaries, which can reveal novel components and mechanisms in higher organisms.

**\*For correspondence:**
d.stojanovski@unimelb.edu.au

**Competing interests:** The authors declare that no competing interests exist.

## Introduction

Mitochondria cannot be created *de novo* and pre-existing mitochondria are used as templates for mitochondrial biogenesis. This genesis requires the ~1500 different mitochondrial proteins to be imported via dynamic translocation machines to one of four subcompartments of the organelle – outer and inner membrane, intermembrane space and matrix (*Chacinska et al., 2009*; *Stojanovski et al., 2012*; *Dolezal et al., 2006*; *Harbauer et al., 2014*; *Neupert and Herrmann, 2007*; *Baker et al., 2014*). The Translocase of the Outer Membrane (TOM) complex is described as the general entry gate to mitochondria and provides a passageway through which precursors can cross the outer membrane. The mitochondrial inner membrane contains two translocase machines that are responsible for the import of a large fraction of the mitochondrial proteome; the Translocase of the Inner Membrane 23 (TIM23) complex and the Translocase of the Inner Membrane 22 (TIM22) complex.

The TIM23 complex typically transports proteins that possess a matrix-targeting N-terminal presequence (*Chacinska et al., 2009*; *Neupert and Herrmann, 2007*; *Wagner et al., 2009*; *Mokranjac and Neupert, 2010*), while the TIM22 complex mediates the inner membrane insertion of multi-transmembrane spanning proteins that contain internal targeting elements (*Chacinska et al., 2009*; *Neupert and Herrmann, 2007*; *Rehling et al., 2004*; *Koehler, 2004*).

**eLife digest** Mitochondria are like tiny bean-shaped "power stations" that provide our cells with the vast majority of the energy that they need. These structures, however, are not self-sufficient and instead rely on proteins and chemicals that are imported from elsewhere in the cell. Two layers of membrane enclose the mitochondria, and transporting proteins across the inner and outer membranes requires large molecular machines embedded within the membranes.

One such complex, the TIM22 complex, organizes tunnel-like carrier proteins that in turn ferry chemicals across the inner membrane to fuel metabolism. The TIM22 complex is vitally important as it allows mitochondria to adapt their metabolism – that is, how and when they generate energy – to match the cell's needs during development. Yet, while the TIM22 complex has been studied extensively in yeast, less is known about how it works in human cells.

Now, Kang et al. have identified a new piece of the human equivalent of the TIM22 machinery, a protein called Tim29, which helps to assemble the TIM22 complex in human cells. Experiments reveal that Tim29 also creates a link between human TIM22 and the TOM complex, a complex that serves as the general entry point through the outer mitochondrial membrane. Sequence analysis revealed that Tim29 is found in other animals, such as chimpanzees and cows, but not in yeast. This suggests that the mitochondrial machinery has changed during evolution.

Kang et al. plan to further investigate how human carrier proteins reach the mitochondria, and exactly how Tim29 helps human TIM22 to cooperate with TOM. Overall, the discovery of Tim29 highlights the importance of looking at mitochondrial machinery across different species in the hope of revealing new components and mechanisms. A future challenge will be to determine how relevant these machines are in human development and diseases.

Substrates of the TIM22 complex include the mitochondrial carrier family, such as the ADP/ATP carrier (AAC) and the phosphate carrier (PiC), and multispanning inner membrane proteins like, Tim17 and Tim23 (subunits of the TIM23 complex) and Tim22 itself (pore forming unit of the TIM22 complex) (*Chacinska et al., 2009*; *Stojanovski et al., 2012*; *Koehler, 2004*; *Sirrenberg et al., 1996*; *Káldi et al., 1998*). In yeast cells, TIM22 is a 300-kDa complex, consisting of four membrane integral subunits, Tim22, Tim54, Tim18 and Sdh3, and a peripheral chaperone complex consisting of the small TIM proteins, Tim9-Tim10-Tim12 (*Adam et al., 1999*; *Gebert et al., 2011*; *Jarosch et al., 1997*, *1996*; *Kerscher et al., 1997*, *2000*; *Koehler et al., 2000*, *1998*; *Kovermann et al., 2002*). The small TIM proteins are a family of intermembrane space chaperones that facilitate the passage of hydrophobic membrane proteins through this aqueous environment. Tim9 and Tim10 form a soluble hexameric complex, but a fraction also interacts with the TIM22 complex via assembly with Tim12 (*Adam et al., 1999*; *Gebert et al., 2008*; *Baud et al., 2007*).

Like yeast, the human TIM22 complex consists of the channel-forming hTim22 protein, along with subunits of the small TIM family, hTim9, hTim10a, and hTim10b (*Mühlenbein et al., 2004*), with hTim10b being the functional homologue of yeast Tim12 (*Koehler et al., 1998*; *Baud et al., 2007*; *Mühlenbein et al., 2004*; *Gentle et al., 2007*). However, homologues of yeast Tim54 or Tim18 are absent in human cells and there is no evidence to indicate that the Sdh3 homologue, SDHC, interacts with the human TIM22 translocase. Thus, the true architecture of the human TIM22 complex remains an open question. Given the many elaborate functions of mitochondria in human cells, including, cell death, metabolism, tumorigenicity and neurodegenerative disorders, we reasoned the composition of the TIM22 complex in human cells is likely different to yeast. This led us to investigate the subunit composition of the human TIM22 complex.

Here we report on the identification of C19orf52 as a novel subunit of the human TIM22 complex, and accordingly rename the protein as Tim29. Tim29 functions in the assembly of hTim22 and consequently the stability of the TIM22 complex. Furthermore, we propose that Tim29 links the TIM22 complex to the general entry gate of mitochondria, the TOM complex. Our findings highlight the importance of analysing mitochondrial protein import across phylogenetic boundaries, since it can reveal novel facilitators of this essential cellular process, in addition to novel mechanisms.

# Results

## The previously uncharacterised protein C19orf52 is a mitochondrial protein and interacts with hTim10b

We sought to investigate the composition of the human TIM22 complex using affinity enrichment approaches. Exogenous expression of hTim22 in cells proved challenging, with the protein displaying a propensity to aggregate (data not shown). However, this was not the case for hTim10b and we therefore generated a stable tetracycline-inducible HEK293T cell line expressing hTim10b[3XFLAG]. The FLAG-tagged protein was exclusively localised to mitochondria when assessed by fluorescence microscopy (*Figure 1A*, upper panel). Blue Native (BN)-PAGE of isolated mitochondria revealed that hTim10b[3XFLAG] formed the same assemblies as the untagged hTim10b protein (assessed by in vitro

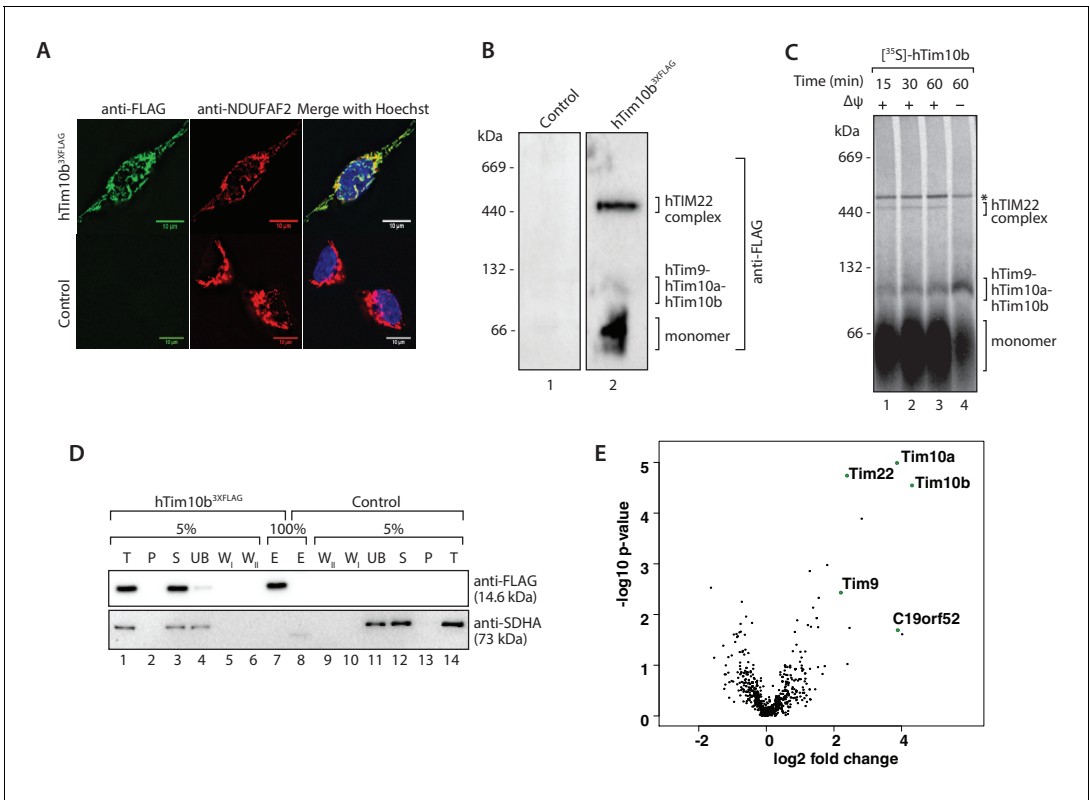

**Figure 1.** The previously uncharacterised protein C19orf52 immunoprecipitates with hTim10b. (**A**) Representative fluorescence images of tetracycline-induced HEK293T cells that contained the Control empty vector (pCDNA5-FRT/TO) or hTim10b[3XFLAG]. Cells were fixed prior to incubation with anti-FLAG (left panel, green; to stain hTim10b[3XFLAG]) and anti-NDUFAF2 (middle panel, red; to visualise the mitochondria). Hoechst stain (far right panel, blue) was used to stain the nucleus. Primary antibodies were counterstained with Alexa Fluor 568 and 488 secondary antibodies prior to microscopy. Scale bar: 10 μm. (**B**) Mitochondria isolated from control cells or cells expressing hTim10b[3XFLAG] were solubilised in digitonin-containing buffer before being analysed by BN-PAGE and immunoblotting using anti-FLAG antibodies. (**C**) [35S]-labelled hTim10b precursor was imported into mitochondria isolated from wild type HeLa cells for the indicated times in the presence or absence of a membrane potential (Δψ). Following import the mitochondria were re-isolated, lysed in digitonin-containing buffer and subjected to blue native electrophoresis and autoradiography. Asterisk (*) indicates non-specific band. (**D**) Mitochondria isolated from control and hTim10b expressing cells were solubilised in digitonin-containing buffer. Mitochondrial lysates were subjected to immunoprecipitation with anti-FLAG resin. Collected fractions were analysed by SDS-PAGE and western blotting using anti-FLAG and anti-SDHA antibodies. T, Total; P, Pellet; S, Supernatant; UB, Unbound; $W_I$ and $W_{II}$, Wash I and II and E, Elution. 5% of the T, P, S, UB, $W_I$ and $W_{II}$ fractions and 100% of the E fraction were loaded for SDS-PAGE analysis. (**E**) Volcano plot showing proteins enriched in hTim10b pull-down versus the empty vector control. All proteins were plotted and each circle represents one protein/gene. The X-axis shows the Log2 fold change of Tim10b interacting partners and Y-axis shows the —log10 of the p-values.

The following source data is available for figure 1:

**Source data 1.** Data from *Figure 1E*.

import of hTim10b into isolated mitochondria), including the hTim9-hTim10a-hTim10b hexamer and a higher molecular weight 450-kDa TIM22 complex (*Figure 1B and C*).

We affinity-purified hTim10b[3XFLAG] and associated proteins using anti-FLAG resin (*Figure 1D*) and a stable cell line containing an empty vector was used as a negative control (since we often observed leaky expression of hTim10b in un-induced cells). A single protein of the expected molecular weight was detected in the eluate fraction by western blotting with FLAG-specific antibodies (*Figure 1D*, lane 7, upper panel). Eluate fractions were then analysed by label-free quantitative mass spectrometry to reveal the interacting partners of hTim10b. Among proteins that co-purified specifically with hTim10b[3XFLAG] were known interacting partners hTim10a, hTim9 and hTim22 (*Figure 1E* and *Figure 1—source data 1*), which suggested the purification of functional TIM22 complexes. The uncharacterised protein, C19orf52, also displayed significant enrichment in the hTim10b[3XFLAG] sample (*Figure 1E* and *Figure 1—source data 1*).

Sequence similarity searches for C19orf52 did not reveal homologs in lower eukaryotes. However, C19orf52 is conserved in metazoan species (*Figure 2—figure supplement 1*). The previously uncharacterised protein is predicted to possess an N-terminal presequence at amino acids 1–16, based on the MitoProt II bioinformatics program (*Claros, 1995*; *Claros and Vincens, 1996*) and a single transmembrane domain spanning amino acids 61–79 (*Figure 2A*). We first performed a mitochondrial in vitro import assay to assess import of the protein into mitochondria. [$^{35}$S]-C19orf52 was synthesised in rabbit reticulocyte lysate and imported into mitochondria isolated from HeLa cells. C19orf52 could be imported into a protease-protected location and in a time-dependent manner, but we did not observe proteolytic processing of the protein (*Figure 2B*, upper panel), as observed for the presequence-containing, pre-ornithine transcarbamylase (pOTC) (*Figure 2B*, lower panel). However, the import of C19orf52 was inhibited under conditions where the membrane potential ($\Delta\psi$) had been dissipated (*Figure 2B*, upper panel, lane 8) in a similar manner to pOTC (*Figure 2B*, lower panel, lane 8), suggesting import via a $\Delta\psi$-dependent import pathway. To confirm the mitochondrial localisation of C19orf52 and define the boundaries of the protein's targeting information, we expressed the following constructs in HeLa cells as C-terminal FLAG-tagged fusion proteins: (i) full length C19orf52 (C19orf52[WT]); (ii) C19orf52 lacking the 16 amino acids that make up the predicted presequence (C19orf52[Δ16]); (iii) the N-terminal 89 amino acids of C19orf52, which includes the predicted presequence and transmembrane domain (aa 61–79) (C19orf52[aa1-89]); (iv) the N-terminal 89 amino acids of C19orf52, but lacking the predicted presequence (C19orf52[aa17-89]) and (v) a N-terminal truncation of C19orf52 containing amino acids 80–260 (C19orf52[aa80-260]). C19orf52 showed exclusive localisation to mitochondria (*Figure 2C*, upper panel), confirming the protein is indeed mitochondrial. C19orf52[Δ16] and C19orf52[aa17-89], which both lack the N-terminal 16 amino acids were both targeted to mitochondria, suggesting that residues 1–16 are dispensable for targeting to the organelle (*Figure 2C*). Expression of C19orf52[aa1-89] also displayed exclusive mitochondrial localisation, while C19orf52[aa80-260] could not be targeted to mitochondria and was localised to the cytosol (*Figure 2C*).

Next we addressed the sub-mitochondrial localisation of C19orf52. We generated a stable tetracycline-inducible HEK293T cell line expressing C19orf52[3XFLAG]. Hypo-osmotic swelling and carbonate extraction treatments were performed on the isolated mitochondria. C19orf52[3XFLAG] was present in isolated mitochondria and only became accessible to external protease when the outer membrane was disrupted (*Figure 2D*, compare lanes 2 and 4), indicating the C-terminal FLAG tag is exposed to the intermembrane space. Outer membrane proteins Bak and Mfn2, Opa1 that is exposed to the intermembrane-space and matrix-located NDUFAF2 served as controls. C19orf52[3XFLAG] was resistant to carbonate treatment, like the membrane integrated Bak and Mfn2, indicating that it is likely an integral membrane protein (*Figure 2E*). In a similar manner, subfractionation and carbonate extraction of mitochondria isolated from cells expressing C19orf52[1-89] revealed that the truncated protein is also an integral membrane protein that exposes it's C-terminal FLAG tag to the intermembrane space (*Figure 2—figure supplement 2A*). Taken together this data indicates that C19orf52 is an integral inner membrane protein with a C-terminal domain exposed to the intermembrane space. Surprisingly, upon subfractionation and carbonate extraction of mitochondria isolated from cells expressing C19orf52[Δ16] and C19orf52[17-89] (*Figure 2—figure supplement 2C and D*) we noticed that these proteins failed to sort properly to the inner membrane, but appeared trapped in the outer membrane, most likely in the TOM complex (indicated by accessibility to protease and presence in both the pellet and supernatant fractions upon carbonate treatment). Thus, although the

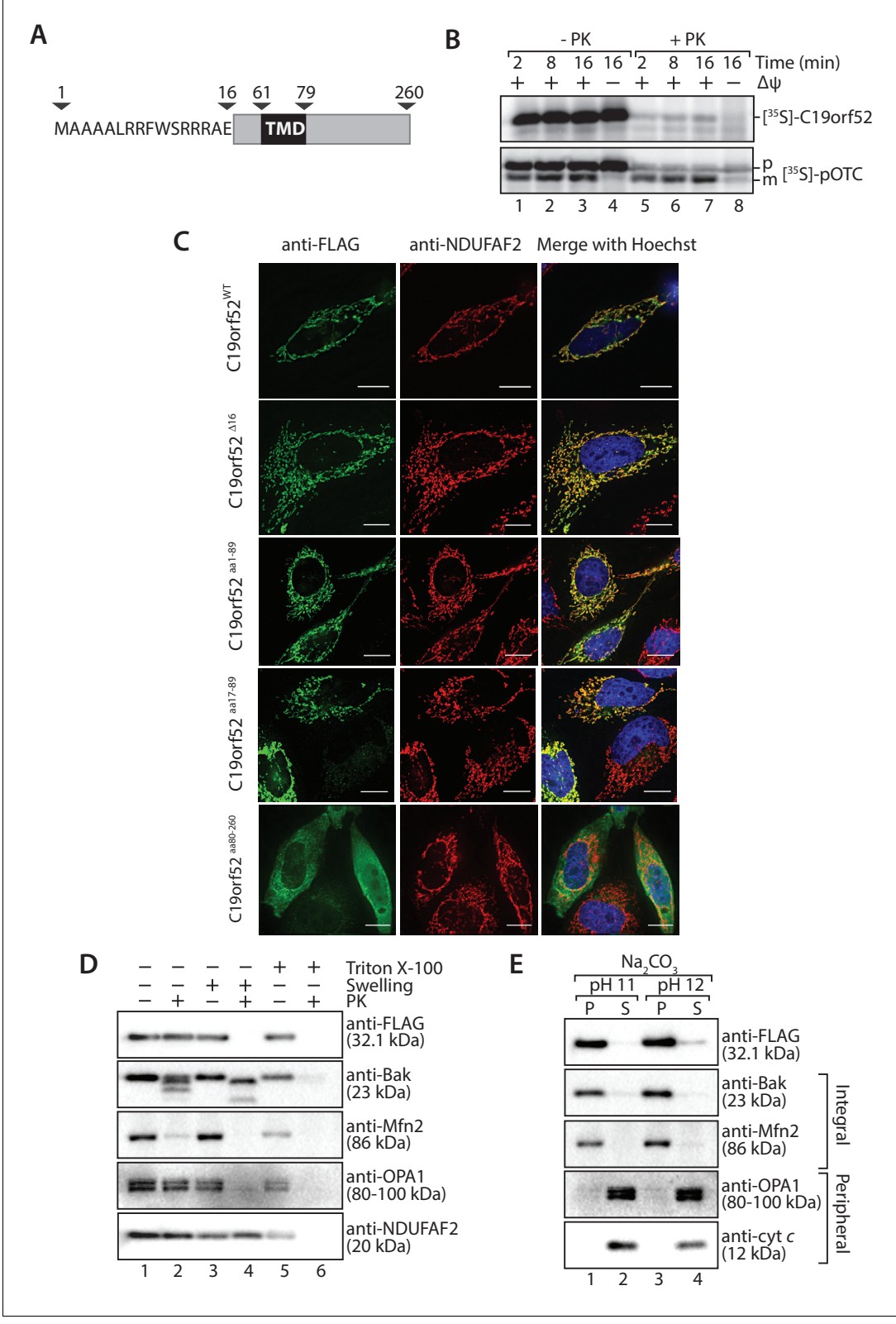

**Figure 2.** C19orf52 is a mitochondrial inner membrane protein. (**A**) Schematic representation of the predicted domain structure for C19orf52 from *Homo sapiens*. (**B**) In vitro import of [$^{35}$S]-labelled C19orf52 and pre-ornithine transcarbamylase, pOTC into mitochondria isolated from HeLa cells for the indicated times in the presence or absence of membrane potential (Δψ). Following import mitochondria were re-isolated and either left untreated

*Figure 2 continued on next page*

*Figure 2 continued*

(lanes 1–4) or treated with Proteinase K (lanes 5–8). Samples were analysed using SDS-PAGE and autoradiography. p, precursor and m, mature. (C) The indicated C19orf52 variants (C19orf52$^{WT}$, C19orf52$^{Δ16}$, C19orf52$^{aa1-89}$, C19orf52$^{aa17-89}$ and C19orf52$^{aa80-260}$) were transiently transfected and expressed as C-terminal 3XFLAG fusions in HeLa cells. Cells were immunostained with anti-FLAG and anti-NDUFAF2 (mitochondria marker) antibodies for visualisation using fluorescence microscopy. Scale bar: 10 μm. (D) Mitochondria were isolated from stable tetracycline-inducible HEK293T cells expressing C19orf52$^{3XFLAG}$. Intact mitochondria (lanes 1 and 2), mitoplasts (generated by hypotonic swelling of the outer membrane, lanes 3 and 4) and solubilised mitochondria (Triton X-100, lanes 5 and 6) were incubated with or without Proteinase K (50 μg/ml) and analysed by SDS-PAGE and western blotting using the indicated antibodies. (E) Mitochondria isolated from C19orf52$^{3XFLAG}$ expressing cells were subjected to alkaline extraction using 100 mM $Na_2CO_3$ (pH 11 and 12). The membrane (P) and soluble (S) fractions were separated by ultra-centrifugation prior to SDS-PAGE and immunoblotting analysis using the indicated antibodies.

The following figure supplements are available for figure 2:

**Figure supplement 1.** C19orf52 is conserved in metazoa.

**Figure supplement 2.** C19orf52 lacking the N-terminal 16 amino acids is not sorted to the inner membrane.

---

first 16 amino acids originally appeared to be dispensable for mitochondrial targeting, closer examination revealed that these residues are indeed crucial for subsequent sorting of C19orf52 to the inner membrane.

## C19orf52 is a subunit of the human TIM22 complex

Next we investigated the association of C19orf52 with the human TIM22 complex. Mitochondria from hTim10b$^{3XFLAG}$ and C19orf52$^{3XFLAG}$ expressing cells were solubilised in digitonin-containing buffer prior to analysis by BN-PAGE and immunodecoration (*Figure 3A*). C19orf52 specific antibodies revealed the presence of a high molecular weight complex of approximately 450-kDa, reminiscent of the TIM22 complex in both control (*Figure 3A*, lane 1) and hTim10b$^{3XFLAG}$ mitochondria (*Figure 3A*, lane 2). We observed a slight delay in the migration of the C19orf52-containing complex in mitochondria isolated from hTim10b$^{3XFLAG}$ expressing cells (*Figure 3A*, lane 2 and 4). This suggested that C19orf52 is located within a hTim10b$^{3XFLAG}$-containing complex and the presence of the FLAG-tag is causing the observed delay in mobility. To interrogate this observation further, we performed antibody-shift analysis of the human TIM22 complex. In this case, [$^{35}$S]-hTim22 was imported into mitochondria isolated from control, hTim10b$^{3XFLAG}$ and C19orf52$^{3XFLAG}$ expressing cells where it assembled into the mature TIM22 complex (*Figure 3B*, lane 1, 4 and 7). Following import, mitochondria were isolated, solubilised in digitonin-containing buffer and incubated with anti-FLAG or anti-SDHA antibodies prior to BN-PAGE. As can be seen (*Figure 3B*, lanes 2, and 8), the FLAG antibodies shifted [$^{35}$S]-hTim22 (hTIM22 complex) in both hTim10b$^{3XFLAG}$ and C19orf52$^{3XFLAG}$ containing mitochondria, suggesting the presence of all three proteins within the same complex. In an alternative approach C19orf52$^{3XFLAG}$ was shown to specifically immunoprecipitate hTim22 (*Figure 3C*), while C19orf52 could be co-immunoprecipitated with hTim10b$^{3XFLAG}$ (*Figure 3D*). As a final source of confirmation immunoprecipitation of C19orf52$^{3XFLAG}$ and subsequent quantitative mass spectrometry-based proteomics analysis revealed the presence of all known TIM22 complex subunits, including hTim22, hTim10b, hTim10a and hTim9 (*Figure 3—source data 1*). These findings confirm that C19orf52 is a *bona fide* subunit of the human TIM22 complex, and represents a novel metazoan specific subunit of the translocation machine. Given that the molecular weight of the protein is 29 kDa and the nomenclature of the mitochondrial protein import machinery (*Pfanner et al., 1996*), we henceforth rename this protein as Tim29 and its gene as *TIMM29*.

## Mitochondria lacking Tim29 are impaired in the import and assembly of TIM22 substrates

To investigate the molecular function of Tim29 at the TIM22 complex we undertook RNA interference studies in HEK293FS cells. Knockdown (KD) of Tim29 was assessed relative to cells transfected

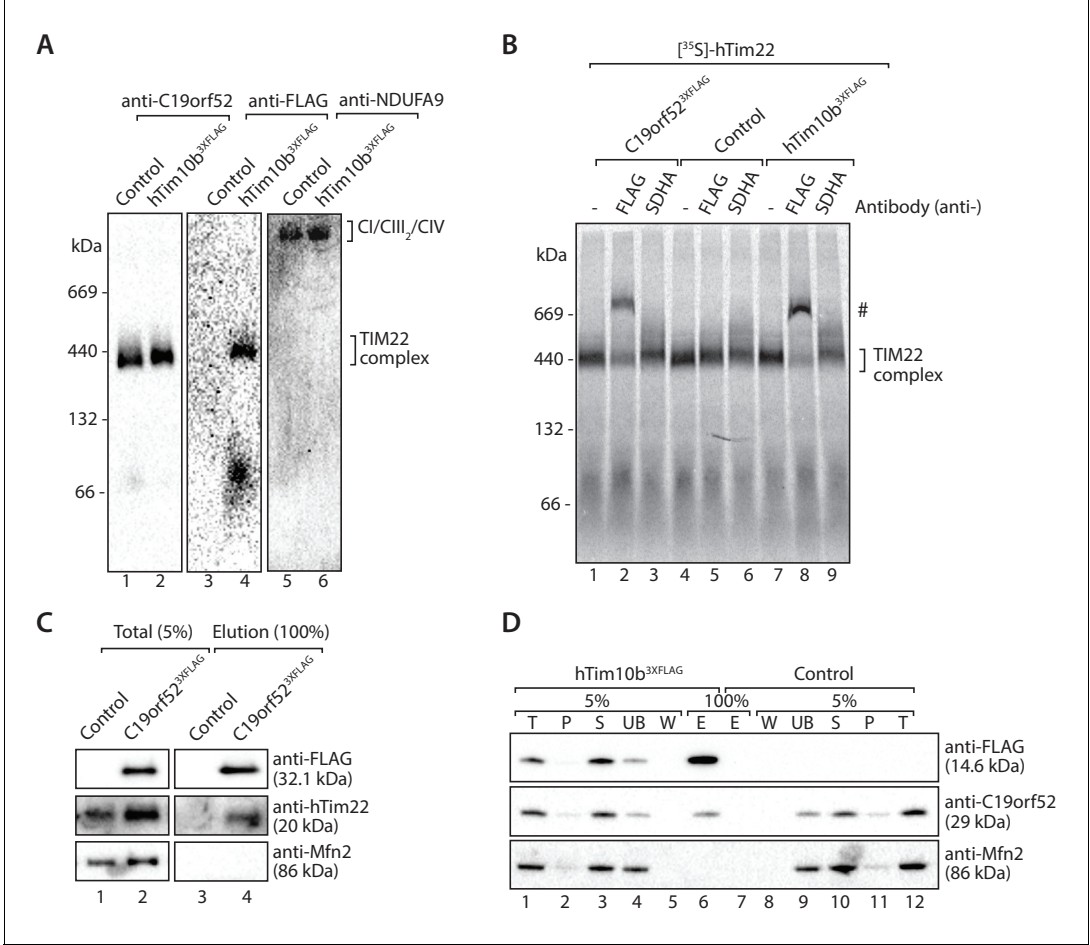

**Figure 3.** C19orf52 is a subunit of the human TIM22 complex. (**A**) Mitochondria isolated from control or hTim10b[3XFLAG]-expressing cells were solubilised in 1% digitonin-containing buffer and analysed by BN-PAGE and western blotting with the indicated antibodies. (**B**) [35S]-labelled hTim22 was imported into mitochondria isolated from control, C19orf52[3XFLAG] or hTim10b[3XFLAG] expressing cells. Following import at 37°C mitochondria were reisolated, solubilised in digitonin-containing buffer and incubated with either anti-FLAG or anti-SDHA antibodies. Samples were separated by BN-PAGE and analysed by autoradiography. # indicates antibody-shifted protein complex. (**C**) Control and C19orf52[3XFLAG] mitochondria were solubilised in digitonin-containing buffer and subjected to immunoprecipitation using anti-FLAG resin. Total (5%) and Elution (100%) fractions were analysed using SDS-PAGE and immunoblotting. (**D**) Digitonin-solubilised mitochondrial lysates from control and hTim10b[3XFLAG] expressing cells were subjected to immunoprecipitation with anti-FLAG resin. Fractions were analysed by SDS-PAGE and western blotting using the indicated antibodies. T, Total; P, Pellet; S, Supernatant; UB, Unbound; W, Wash and E, Elution.

The following source data is available for figure 3:

**Source data 1.** C19orf52 immunoprecipitates TIM22 complex subunits.

with a non-targeting control and we also performed KD of hTim22, which allowed us to monitor the implications of shutting down the carrier pathway (*Figure 4—figure supplement 1A and B*). To address the global impact of Tim29 and hTim22 KD on cell health and viability we monitored the impact of Tim29 and hTim22 KD on mitochondrial respiration, cell proliferation and cell viability. We measured mitochondrial oxygen consumption rates in intact cells by high-resolution respirometry to determine if the KD of Tim29 and hTim22 affects mitochondrial metabolism. Knockdown of Tim29 significantly reduced basal mitochondrial oxygen consumption to 65% ($p < 0.05$) compared to cells transfected with a control scrambled siRNA (*Figure 4—figure supplement 1C*). This reduction in basal oxygen consumption was similar to that observed in hTim22 KD cells (61% residual respiration, $p < 0.05$). Following addition of FCCP, the maximal mitochondrial oxygen consumption rate was also significantly lower in Tim29 KD cells compared to control cells (88% residual respiration, $p < 0.05$) and

in hTim22 knockdown cells compared to the control (63%, p<0.05). KD of both Tim29 and hTim22 did not have any significant impact on cell proliferation (*Figure 4—figure supplement 1D*), or cell viability (*Figure 4—figure supplement 1E*).

Next we assessed the impact of KD of Tim29 on TIM22 complex substrates at the steady state protein level. As can be seen, Tim29 (*Figure 4A*) and hTim22 (*Figure 4B*) were depleted from mitochondria while the control protein Mfn2 remained unaffected (*Figure 4A and B*, bottom panels). Specific depletion of TIM22 complex substrates, including hTim23, ANT3 and the glutamate carrier was observed in both Tim29 and hTim22 depleted mitochondria (quantifications shown in *Figure 4A and B*, lower panels). The depletion of Tim29 also led to a significant reduction in the levels of hTim22 (*Figure 4A*), while the lack of hTim22 had no obvious effect on the levels of Tim29, indicating Tim29 is imported in a TIM22-independent manner (*Figure 4B*). BN-PAGE analysis also revealed that depletion of Tim29 caused a reduction in the assembly of ANT3 and hTIM23 complexes (*Figure 4C*), with the latter causing a reduction in the steady state level of TIM23 complex substrates (COXIV, NDUFV2 and NDUFV1) (*Figure 4—figure supplement 2A*) and import of [$^{35}$S]-NDUFV1 and [$^{35}$S]-NDUFV2 (*Figure 4—figure supplement 2B and C*).

## Tim29 is required for the assembly of hTim22

The decrease in Tim29 protein levels also revealed a reduction in the levels of the endogenous TIM22 complex on BN-PAGE (*Figure 4D*). This observation and the decrease of endogenous TIM22 complex substrates in Tim29 KD cells, prompted us to address if this decrease was due to: (i) an active role of Tim29 in the import of substrates into the inner membrane, or (ii) a role of Tim29 in the biogenesis of the TIM22 complex itself. To address this, we investigated the in vitro import and assembly of two well-established substrates of the TIM22 pathway, ANT1 and hTim23. While the assembly of ANT1 was reduced in the absence of hTim22 at 60 min (*Figure 5A*; right panel), assembly of the protein proceeded like WT in the absence of Tim29 (*Figure 5A*; left panel). In contrast, the assembly of hTim23 was mildly reduced in mitochondria depleted of either Tim29 or hTim22 (*Figure 5B*). We next addressed the assembly of the TIM22 complex itself using the radiolabelled hTim22. In mitochondria isolated from control cells, [$^{35}$S]-hTim22 efficiently assembled into the 450-kDa TIM22 complex (*Figure 6A and B*, lanes 1–4), while there was a clear assembly defect into the TIM22 complex in mitochondria isolated from Tim29 KD cells (*Figure 6A*, lanes 5–8). The amount of assembled TIM22 complex in Tim29 KD mitochondria was lower than that in the control mitochondria by 53%, after a 60 min-import. To support this observation, we examined the difference in hTim22 protein assembly in control and Tim29 KD mitochondria using a two-tailed Student's *t*-test. We obtained a *t*[2] = 6.02 (p<0.05), demonstrating a significant reduction in the assembly of hTim22 into the mature complex in Tim29 KD mitochondria. On the contrary, we consistently detected a moderate increase (~20%) in assembled TIM22 complex in hTim22 KD mitochondria after a 60 min-import compared to control mitochondria (*t*[2] = −5.07, p<0.05).

In the absence of Tim29 we did observe a minor decrease in the import kinetics of [$^{35}$S]-hTim22 by SDS-PAGE (*Figure 7*), however, following treatment of isolated mitochondria with sodium carbonate it was apparent that [$^{35}$S]-hTim22 protein efficiently integrated into the inner membrane in mitochondria isolated from both Tim29 and hTim22 KD cells (*Figure 7C and D*). Based on these observations, we conclude that Tim29 is required for the assembly of the hTim22 protein into the TIM22 complex.

## Tim29 links the hTIM22 complex to the TOM complex

Coupling of the TOM and TIM23 complexes in yeast mitochondria is believed to enhance the import efficiency of presequence-containing precursors (*Chacinska et al., 2005*, *2003*; *Waegemann et al., 2015*; *Mokranjac et al., 2009*; *Popov-Čeleketić et al., 2008*). More recently, coupling between the TOM and SAM complexes of the outer membrane in yeast mitochondria has been shown to be important for the biogenesis of β-barrel proteins (*Wenz et al., 2015*; *Qiu et al., 2013*). Interestingly, no interaction has ever been reported between the TOM and TIM22 translocases in yeast mitochondria. Rather, the carrier pathway is described as a series of five stages, with stage 3 encompassing the transfer of substrates between the TOM and TIM22 complexes by the soluble intermembrane space Tim9-Tim10 complex (*Chacinska et al., 2009*; *Rehling et al., 2004*; *Ryan et al., 1999*).

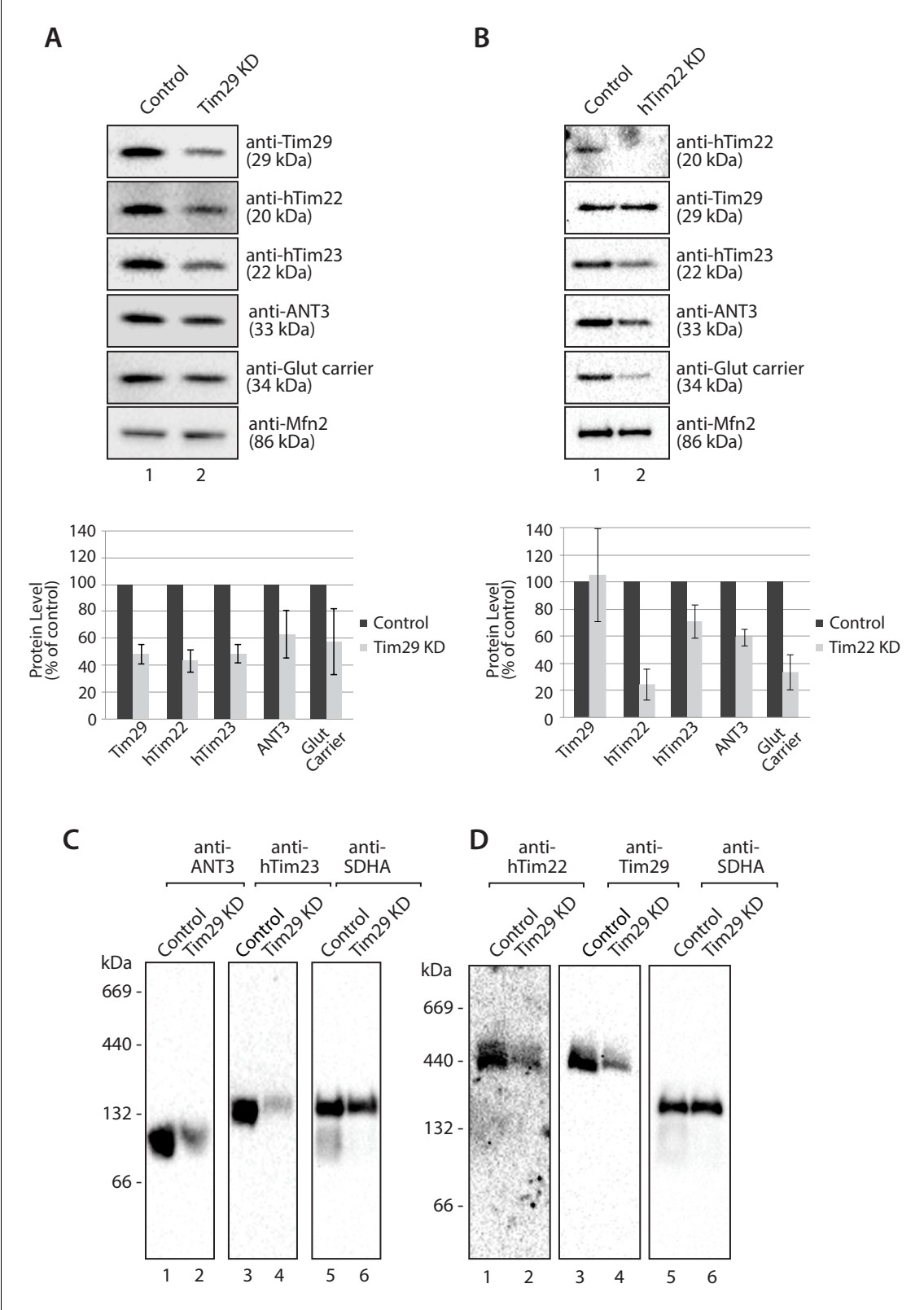

**Figure 4.** Knockdown of Tim29 reduces the steady state protein levels of TIM22 substrates. (**A**) HEK293FS cells were transfected with scrambled (control) or Tim29 siRNA targets for 96 hr. Following knock-down mitochondria were isolated from cells and analysed using SDS-PAGE and western blotting analysis with the indicated antibodies. Protein levels were quantified and normalised against the loading control, Mfn2. The amount of each protein in control cells was set to 100%. Data are expressed as mean ± SD, n = 3. (**B**) Mitochondria were isolated from control cells (scrambled siRNA)

*Figure 4 continued on next page*

*Figure 4 continued*

or cells transfected with hTim22 siRNA target for 96 hr. Mitochondrial proteins were subjected to SDS-PAGE and immunoblotting using the indicated antibodies and quantified as described above (**C & D**) Mitochondria were isolated from control or Tim29 knock down (KD) cells and solubilised in 1% digitonin-containing buffer. Protein complexes were analysed by BN-PAGE and western blotting.

The following figure supplements are available for figure 4:

**Figure supplement 1.** Cellular consequences of Tim29 depletion.

**Figure supplement 2.** TIM23 complex substrates are reduced in cells depleted of Tim29.

Given that Tim29 exposes a domain into the intermembrane space, we reasoned it could have a role in linking the TIM22 complex to the TOM complex in mammalian mitochondria. Mitochondria were isolated from control and Tim29$^{3XFLAG}$ expressing cells, solubilised in digitonin and incubated with anti-FLAG resin for immunoprecipitation of the FLAG-tagged proteins and associated partners. As expected, we observed co-elution of hTim9 with Tim29$^{3XFLAG}$. Interestingly, we observed a fraction of both hTom40 and hTom22 co-eluting with Tim29$^{3XFLAG}$, suggesting a co-operation between Tim29 and the TOM complex (*Figure 8A*). To confirm this interaction, we imported radiolabelled versions of hTom40, hTom22 and hTim9 (positive control) into mitochondria isolated from control and Tim29$^{3XFLAG}$ expressing cells and following import subjected the isolated mitochondria to immunoprecipitation with anti-FLAG resin (*Figure 8B*). Indeed a fraction of both hTom40 and hTom22 were observed in the Tim29 elution fraction (*Figure 8B*, middle and lower panels), like that of the control protein hTim9 (*Figure 8B*, upper panel).

If the interaction we observed between TIM22 and TOM was authentic, it should be possible to perform the reverse experiment, i.e. copurification of the TIM22 complex with a tagged TOM complex. We generated an N-terminally-tagged variant of TOM22, which localised to mitochondria and assembles into the TOM complex (*Figure 8—figure supplement 1*). Upon solubilisation of the $^{3XFLAG}$hTom22 mitochondria and immunoprecipitation we observed co-purification of hTom40 and hTom70 (*Figure 8C*) as expected. We also observed a fraction of hTim9, Tim29 and hTim22 co-purifying with the TOM complex, while numerous controls of the outer membrane (hFis1), intermembrane space (cytochrome *c*), inner membrane (OPA1, hTim23, Glutamate carrier, COXIV) and matrix (NDUFV1, NDUFV2) did not co-elute with hTom22 (*Figure 8C*). Given that the TIM22 complex was not purified in great amounts, the data suggest the interaction between TOM and TIM22 complexes is labile in the absence of precursor protein, or under the conditions employed for the immunoprecipitation.

To explore the relationship between TOM and TIM22 further, we performed chemical cross-linking with dithiobis (succinimidylpropionate) (DSP), an amino-group-specific homobifunctional cross-linker that is cleaved by reducing agents. Mitochondria isolated from control and Tim29$^{3XFLAG}$ expressing cells were incubated with DSP, solubilised in SDS to dissociate non-covalent interactions and then incubated with anti-FLAG resin for purification of Tim29$^{3XFLAG}$-containing species. Eluate fractions were treated under reducing conditions in order to liberate cross-linked species. Interestingly, we found hTom40 cross-linked to Tim29 (*Figure 9A and B*), but not hTom22 (*Figure 9B*, third panel from top). We also observed crosslinks between Tim29 and hTim9, but not hTim22 (most likely due to the lack of primary amines in the predicted Tim29 transmembrane domain, which are essential for the amine-reactive crosslinker DSP) (*Figure 9A*). We next performed cross-linking using mitochondria isolated from cells expressing hTim10b$^{3XFLAG}$ and although we observed cross-links of hTim10b with both hTim9 and Tim29 we did not observe any crosslinks with hTom40 (*Figure 9C*). Taken together, these results suggest that Tim29 acts as a bridge between the TIM22 and TOM complexes.

## Discussion

In this study, we have identified C19orf52 as a new subunit of the human TIM22 complex, and have retitled this protein Tim29. While the yeast TIM22 complex and import pathway have been extensively characterised, the human TIM22 complex has remained ill defined. Conservation of members

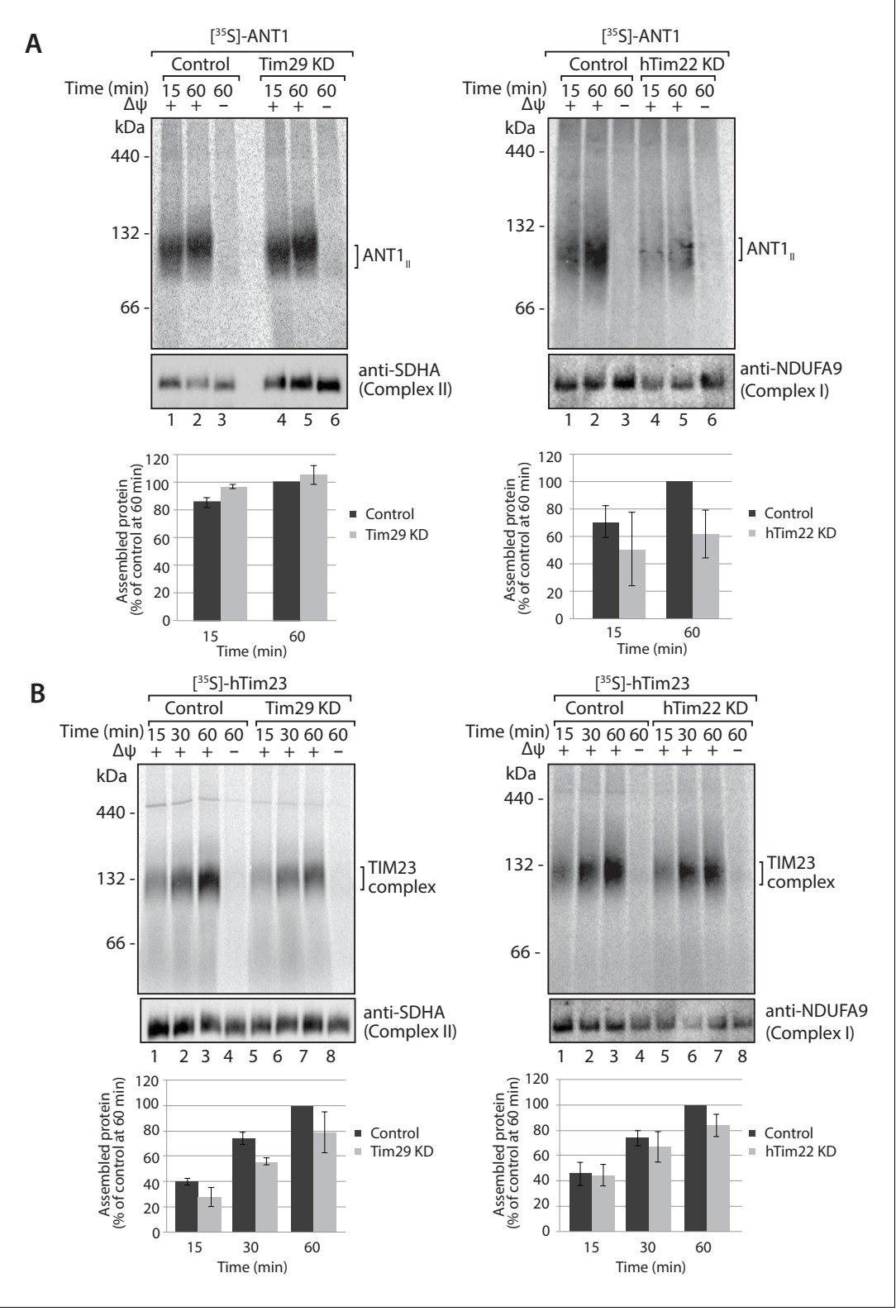

**Figure 5.** Knockdown of Tim29 influences the assembly of the TIM23 complex, but not ANT1. (**A**) [$^{35}$S]-labelled ANT1 was imported into mitochondria isolated from control or Tim29 (left panel) or hTim22 (right panel) knock-down (KD) cells for the indicated times in the presence or absence of membrane potential (Δψ). Re-isolated mitochondria were treated with 50 μg/ml of Proteinase K prior to solubilisation in digitonin-containing buffer. Mitochondrial lysates were analysed by BN-PAGE and autoradiography. Assembled ANT1 was quantified from

*Figure 5 continued on next page*

*Figure 5 continued*
three independent experiments. The amount of assembled ANT1 in control mitochondria at 60 min was set to
100%. Data are shown as mean ± SD (n = 3). (**B**) Import of [35S]-hTim23 was performed and quantified as described
above.

of the human TIM22 complex (hTim22 and small TIM proteins) based on sequence homology was
originally published approximately 17 years ago (*Bauer et al., 1999*). Advances and accessibility in
techniques for purification of native complexes and mass spectrometry have now provided us with
an opportunity to unravel the true subunit composition of the human complex. Our work brings the
number of known subunits of the human TIM22 complex to five: hTim22 (*Sirrenberg et al., 1996*;
*Bauer et al., 1999*), hTim9, hTim10a, hTim10b (*Mühlenbein et al., 2004*; *Gentle et al., 2007*;
*Bauer et al., 1999*) and Tim29 (this study). Interestingly, our immunoprecipitations and mass spec-
trometry data for both hTim10b[3XFLAG] and C19orf52[3XFLAG] did not reveal the presence of SDHC,
suggesting the human protein is not a subunit of the human TIM22 complex, at least under the con-
ditions in which we perform the immunoprecipitation experiments. *Gebert et al. (2011)* showed
that the assembly of yeast Sdh3 at both Complex II and TIM22 is influenced by the formation of

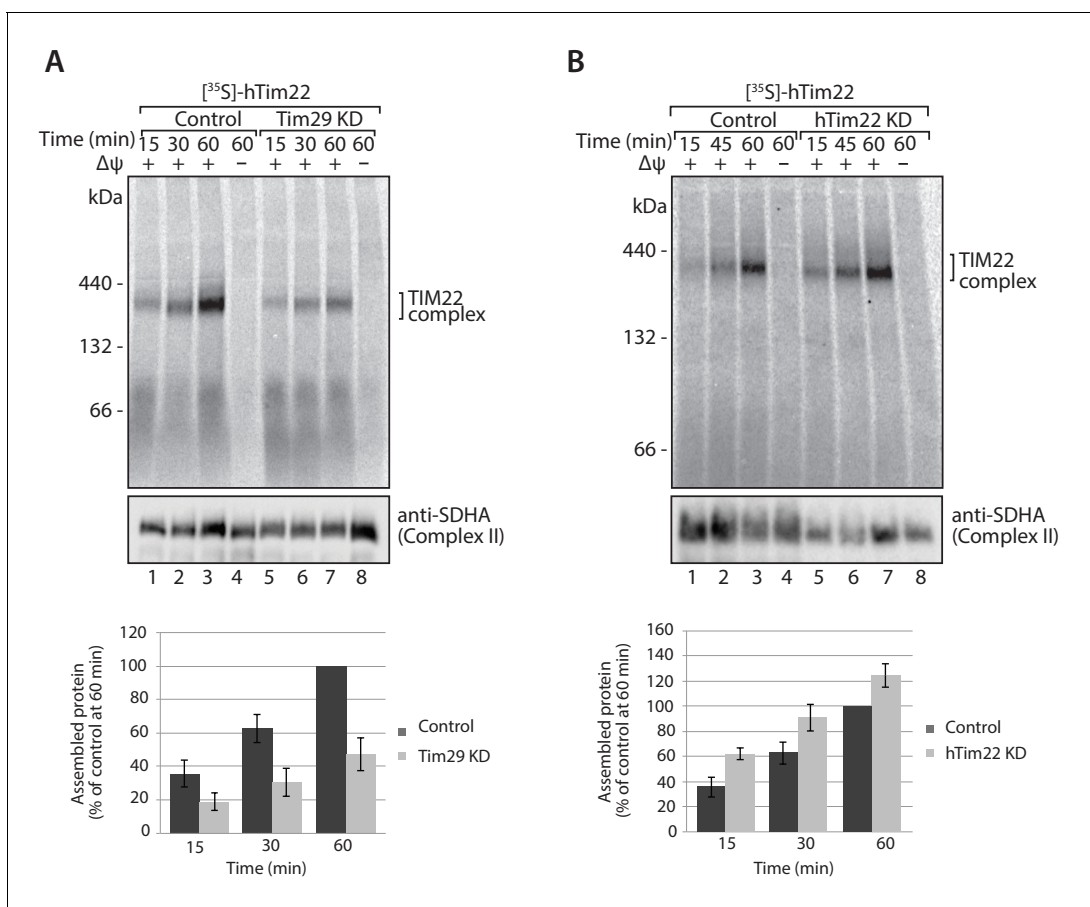

**Figure 6.** Tim29 is required for the assembly of hTim22 into the mature TIM22 complex. (**A**) [35S]-hTim22 was imported into mitochondria isolated from
control (scrambled siRNA) or Tim29 knock-down (KD) cells for the indicated times in the presence or absence of membrane potential (Δψ). Re-isolated
mitochondria were treated with 50 μg/ml of Proteinase K prior to solubilisation in digitonin-containing buffer. Mitochondrial lysates were analysed by
BN-PAGE and autoradiography. Assembled hTim22 was quantified from three independent experiments. The amount of assembled hTim22 in control
mitochondria at 60 min was set to 100%. Data are shown as mean ± SD (n = 3). (**B**) [35S]-hTim22 was imported into mitochondria isolated from control
(scrambled siRNA) or hTim22 knock-down cells and treated as described in (**A**). The amount of assembled hTim22 in control mitochondria at 60 min
was set to 100%. Data are represented as mean ± SD (n = 3).

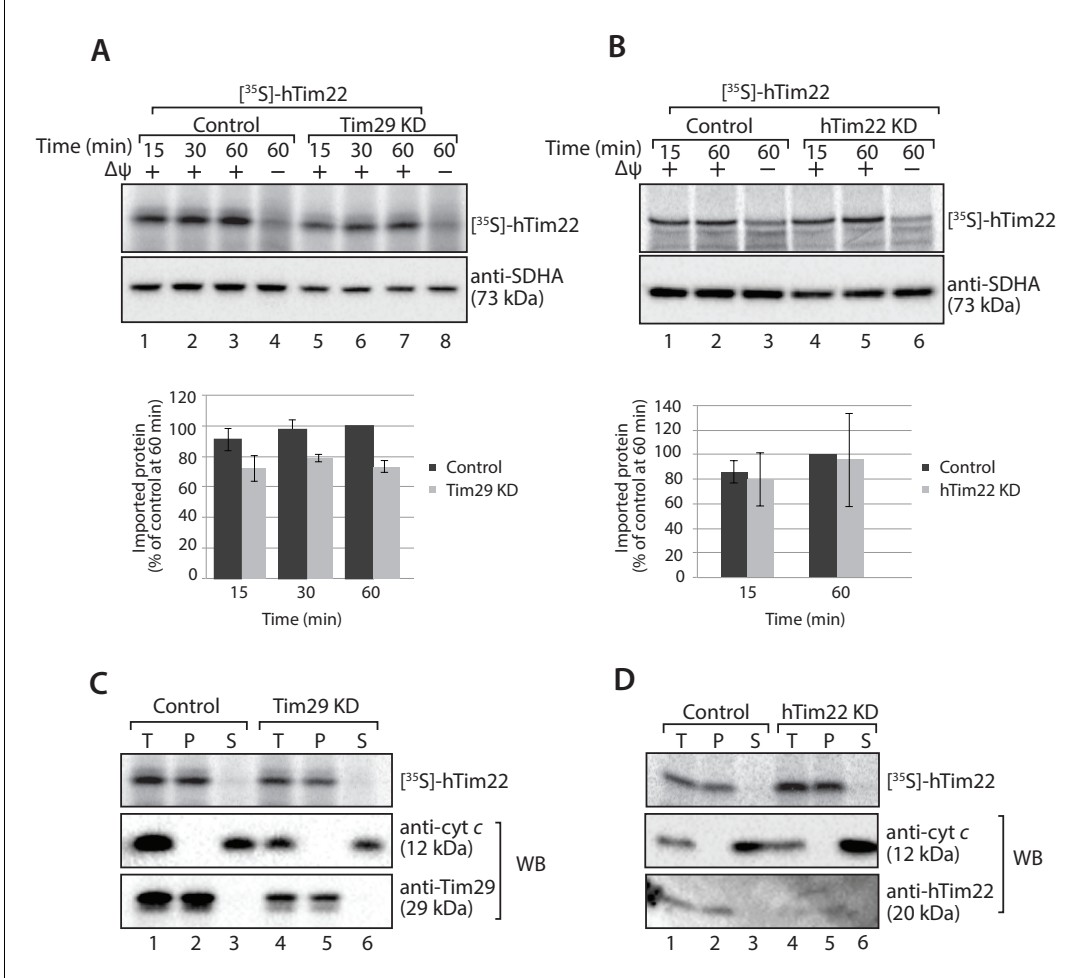

**Figure 7.** hTim22 is integrated into the inner membrane in the absence of Tim29. (**A** and **B**) [35S]-hTim22 was imported into mitochondria isolated from control (scrambled siRNA), and Tim29 (**A**) or hTim22 (**B**) knock-down cells for the indicated times in the presence or absence of membrane potential (Δψ) and analysed by SDS-PAGE. The import efficiency of [35S]-hTim22 into control or Tim29/hTim22 knock-down mitochondria was quantified (mean ± SD, n = 3) and the yield at the longest import time into the control mitochondria was set to 100%. (**C** and **D**) [35S]-hTim22 was imported into mitochondria isolated from control (scrambled siRNA), and Tim29 (**C**) or hTim22 (**D**) knock-down cells for 45 min. Mitochondria were isolated and treated with sodium carbonate and separated into a membrane integrated (pellet, P) and peripheral membrane protein (supernatant, S). WB, indicates western blot.

different sub-complexes with the highly homologous Sdh4 and Tim18. The Sdh3-Sdh4 and Sdh3-Tim18 subcomplexes subsequently deliver the protein to either Complex II or TIM22, respectively. It is feasible to suggest that the lack of Tim18 in human mitochondria could prevent the association of SDHC to the hTIM22 complex.

Tim29 was a previously uncharacterised protein, however the MitoCarta 2.0 inventory indicated the protein was a mitochondrial protein with a wide tissue distribution (*Calvo et al., 2016*). Mitochondrial subfractionation and carbonate extraction confirmed that Tim29 is an integral protein of the mitochondrial inner membrane. Given that there is only one predicted transmembrane region, we propose Tim29 is inserted into the mitochondrial inner membrane via this single transmembrane segment, with its amino-terminus facing the matrix and carboxyl-terminus exposed to the intermembrane space. Interestingly, we confirmed that targeting and proper sorting of Tim29 to the inner membrane relies on the first 89 amino acids of the protein. Although fluorescence microscopy suggested that constructs lacking the first 16 amino acids (Tim29$^{\triangle 16}$ and Tim29$^{aa17-89}$) could be targeted to mitochondria, closer investigation using mitochondrial subfractionation and carbonate extraction reveals these proteins failed to sort to the inner membrane. Given this observation and

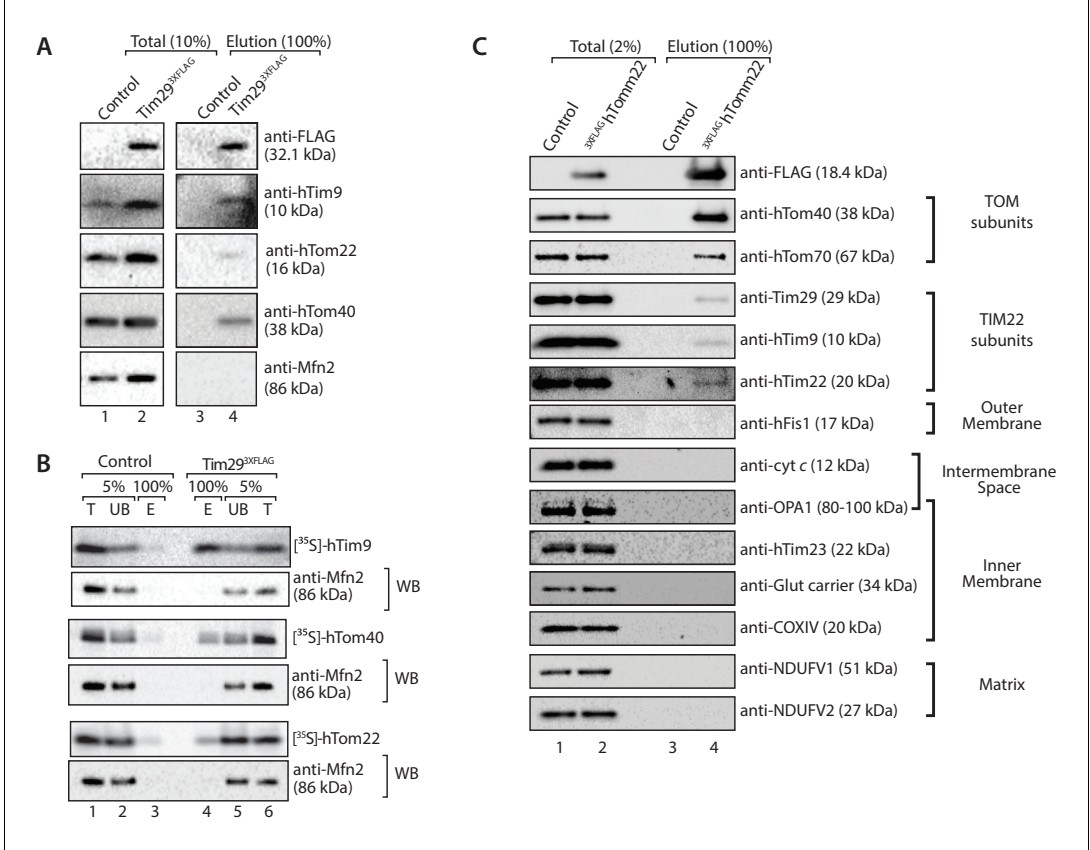

**Figure 8.** Tim29 couples the TIM22 complex to the TOM complex. (**A**) Mitochondria were isolated from control and Tim29[3XFLAG] expressing cells and were solubilised in 0.5% digitonin-containing buffer prior to immunoprecipitation with anti-FLAG resin. Total (10%) and Elution (100%) fractions were collected and analysed using SDS-PAGE and immunoblotting using the indicated antibodies. (**B**) [35S]-hTim9, [35S]-hTom40 and [35S]-hTom22 were imported into mitochondria isolated from control cells or cells expressing Tim29[3XFLAG] for 60 min. Following import samples were solubilised in 0.5% digitonin-containing buffer prior to immunoprecipitation with anti-FLAG resin. Total (T; 5%), Unbound (UB; 5%) and Elution (E; 100%) fractions were separated on SDS-PAGE followed by autoradiography and western blotting using Mfn2 antibody. (**C**) Control and [3xFLAG]hTom22-harbouring mitochondria were lysed in 0.5% digitonin-containing buffer. Mitochondrial extracts were subjected to immunoprecipitation and were separated on SDS-PAGE prior to immunoblotting analysis. Total, 2% and Elution, 100%.

The following figure supplement is available for figure 8:

**Figure supplement 1.** [3XFLAG]hTom22 localises to mitochondria and assembles within the TOM complex.

that the knockdown of hTim22 did not influence the levels of Tim29, the protein is most likely imported via the TIM23 complex, however this needs to be experimentally verified.

Knockdown of Tim29 resulted in a reduction in the steady-state levels of ANT, but import and assembly kinetics of radiolabelled ANT were similar to control mitochondria. In contrast, hTim23, which is also a substrate for the hTIM22 complex showed an assembly defect both in vitro and at the steady state level. Such a scenario has previously been reported for the yeast TIM22 complex, with deletion of Tim18 differentially affecting the import kinetics of the AAC and Tim23 (*Kerscher et al., 2000*; *Koehler et al., 2000*). It is possible that depletion of Tim29 affects the activity of the human TIM23 complex by a more direct mechanism, whereas the import of ANT only requires the TIM22 complex and the residual complex remaining following knockdown is enough to import and assemble the small amounts of radiolabelled protein used in the in vitro import assay.

The depletion of Tim29 also caused a reduction in the levels of the TIM22 complex when assessed by BN-PAGE. Given that hTim22 is integrated into the membrane in mitochondria isolated from the Tim29 KD cells, we conclude that Tim29 is required for the assembly of hTim22 into the TIM22 complex. In addition to this role our data indicate that Tim29 facilitates cooperation between

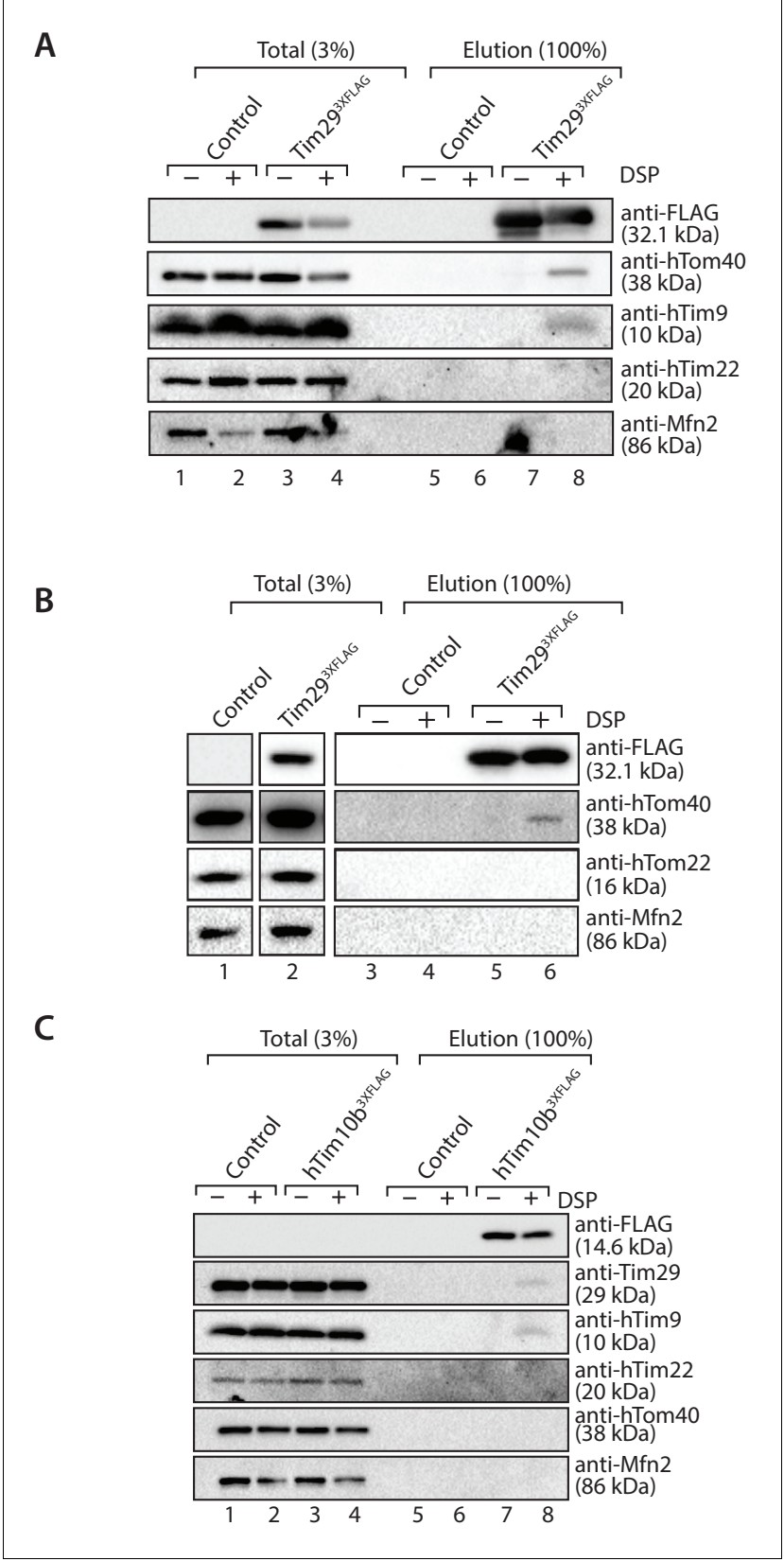

**Figure 9.** Tim29 interacts with hTom40. (**A** and **B**) Mitochondria isolated from control and Tim29³ˣᶠᴸᴬᴳ expressing cells were incubated with 0.2 mM DSP. Upon quenching of excess cross-linker, mitochondria were solubilised in

*Figure 9 continued on next page*

*Figure 9 continued*

SDS-containing buffer and subjected to immunoprecipitation using anti-FLAG resin. Cross-linked species were cleaved using the Laemmli buffer. Total (3%) and Elution (100%) fractions were analysed using SDS-PAGE and western blotting with the indicated antibodies. (C) Mitochondria isolated from control and hTim10b[3XFLAG] expressing cells were treated as described for **A** and **B**.

the TIM22 and TOM complexes, by directly interacting with hTom40. Cooperation between the TOM and TIM23 complexes is well established in yeast cells (*Chacinska et al., 2005*, *2003*; *Waegemann et al., 2015*; *Mokranjac et al., 2009*; *Popov-Čeleketić et al., 2008*; *Chacinska et al., 2010*), however a similar interaction between the TOM and TIM22 complex has not been reported thus far. We have captured an interaction between the TOM and TIM22 complexes *in organello* using both Tim29 and hTom22 as bait. Furthermore, using chemical crosslinking we narrowed down specific interactions between: (i) Tim29 and hTim9; (ii) Tim29 and hTim10b; hTim10b and hTim9; and Tim29 and hTom40. We were puzzled to see an interaction between Tim29 and hTom40, since in yeast TIM23 interacts with TOM via Tom22, which exposes a domain into the intermembrane space (*Waegemann et al., 2015*; *Chacinska et al., 2010*). However, Shiota and colleagues (2015) recently showed that the intermembrane space chaperone, Tim10 interacts with an N-terminal region of Tom40. This suggests that the N-terminal region of Tom40 extends through the β-barrel pore into the intermembrane space (*Shiota et al., 2015*), and such a mechanism could be mediating the interaction between hTom40 and the intermembrane space localised region of Tim29.

In yeast the intermembrane space chaperones Tim9-Tim10 passage carrier precursors through the aqueous intermembrane space delivering them from TOM to TIM22 (*Chacinska et al., 2009*; *Rehling et al., 2004*; *Ryan et al., 1999*). Why would a TOM-TIM22 interaction be necessary in human mitochondria? The answer could lie with the nature of the small TIM chaperones within mammalian mitochondria. Early studies by Mühlenbein and colleagues (2004) suggested that human mitochondria lack soluble Tim9-Tim10 complexes, but rather these proteins are more tightly associated with the mitochondrial inner membrane. Thus, even though the function of the small TIM proteins has been conserved through evolution, the import of carrier proteins into mammalian mitochondria is efficient even in the absence of soluble Tim9-Tim10 complexes. Therefore, it is likely that other mechanisms have evolved that serve to prevent the hydrophobic carrier proteins from aggregating in the intermembrane space. A close association between the TOM and TIM22 complexes could be one such mechanism. Our data represents a major conceptual advance and challenges the common dogma that carrier proteins are transported across the mitochondrial intermembrane space via soluble intermediates. Further investigation is necessary to tease out the precise role of the TOM-TIM22 interaction and how it is influenced by the presence of precursor protein.

In summary, by assessing the subunit composition of the human TIM22 complex, we have identified Tim29 and uncovered a novel interaction between the TIM22 and TOM complexes. This suggests that novel mechanisms are overseeing the import of carrier proteins into human mitochondria. Interestingly, Tim29 (C19orf52) was recently shown to be up-regulated in primary hepatocellular carcinoma cells (*Xing et al., 2015a*, *2015b*). Indeed, numerous human pathologies have been linked to protein import dysfunction (*Sokol et al., 2014*; *Xu et al., 2010*; *Sinha et al., 2014*; *Di Fonzo et al., 2009*; *Koehler et al., 1999*; *Roesch et al., 2002*; *Sankala et al., 2011*; *Davey et al., 2006*). Thus, examining protein import machineries and pathways in human cells is essential and will pave the way to deeper understanding of how mitochondrial protein import influences cell viability, physiology and ultimately human pathologies.

## Materials and methods

### Cell culture and generation of stable cell lines

Cell lines used in this work were Flp-IN T-REx 293 (purchased from Thermo Fisher Scientific Australia; RRID:CVCL_U427); FreeStyle 293-F (HEK293FS; purchased from (Thermo Fisher

Scientific; RRID:CVCL_D603) and HeLa cells (RRID:CVCL_0030); kindly provided by Prof. Paul Gleeson, The University of Melbourne). All cell lines were mycoplasma negative based on screening using MycoAlert Mycoplasma Detection Kit (Lonza). Cell lines were authenticated using short-tandem repeat (STR) typing using the Cell Line Authentication Service at the Garvan Institute of Medical Research (Sydney, Australia). Cells were grown at 37°C and 5% $CO_2$ in Dulbecco's modified Eagle's medium (DMEM; Thermo Fisher Scientific) supplemented with 5–10% (v/v) fetal bovine serum (FBS; in vitro Technologies, Australia). Stable tetracycline-inducible cell lines (Tim10b[3XFLAG] and Tim29[3XFLAG]) were generated using the Flp-IN T-REx System (Thermo Fisher Scientific). Briefly, Flp-IN T-REx HEK293T cells were co-transfected with the plasmid pOG44 (allowing expression of Flp-recombinase) and the desired open reading frame cloned into pCDNA5 /FRT/TO (modified to contain a 3X FLAG tag). We simultaneously generated a control cell line transfected with empty pCDNA5 /FRT/ TO. Transfected cells were placed under selection with hygromycin at a concentration of 200 µg/ml. Cells were incubated until individual colonies appeared and these were selected and expanded and screened for protein expression by the addition of tetracycline at 1 µg/ml. Cells were cultured in with 5–10% (v/v) tetracycline-reduced FBS (Clontech).

## siRNA and transfection

For ectopic expression and generation of stable cells Lipofectamine 2000 was used according to the manufacturer's guidelines (Thermo Fischer Scientific). Custom siRNA oligonucleotides were purchased from Sigma-Adrich. The following sequences and concentrations were utilised: Tim29 (5′ GGCUCUUCGAUGAGAAGUA 3′, 10 nM) and hTim22 (5′ CCAUUGUGGGAGCCAUGUU 3′, 10 nM). For RNA interference HEK293FS cells were (9 × 10$^5$ cells) were transfected with DharmaFECT1 transfection reagent according to the manufacturer's instructions (Dharmacon). Cells were analysed after 72 or 96 hr post-transfection for protein knock-down.

## Immunofluorescence and microscopy

For immunofluorescence, cells grown on coverslips were fixed by incubation with 4% (w/v) paraformaldehyde solution (Sigma-Aldrich) for 10 min at room temperature and permeabilised by incubation with PBS containing 0.2% (v/v) Triton X-100. This was followed by incubation with the desired primary antibody made up in 3% (w/v) bovine serum albumin, 0.2% (v/v) Triton X-100 in PBS for 1 hr. Following washing in 0.2% (v/v) TX-100 cells were incubated with secondary antibody anti-mouse Alexa Fluor 488 (RRID:AB_2534069) or anti-rabbit Alexa Fluor 568 (RRID:AB_2534078; Molecular Probes). For nuclear staining, cells were incubated with 0.01 mg/ml Hoechst 33258 (Sigma-Aldrich) in PBS for 5 min following incubation with secondary antibodies. Cells were analysed using Deltavision DV Elite Imaging System (Applied Precision) and then deconvolved images were analysed using ImageJ.

## Mitochondrial isolation and mitochondrial procedures

Isolation of mitochondria from tissue culture cells was performed by differential centrifugation as described previously (*Johnston et al., 2002*). For mitochondrial sub-fractionation experiments mitochondrial pellets (50 µg of mitochondrial protein) were resuspended in the isolation buffer (20 mM HEPES-KOH pH 7.4, 70 mM sucrose, 220 mM Mannitol, 1 mM EDTA), swelling buffer (10 mM HEPES-KOH pH 7.4) or solubilisation buffer (0.5% [v/v] Triton-X-100). Samples were split and either left untreated or treated with proteinase K (50 µg/ml) for 10 min on ice. For sodium carbonate extraction, mitochondrial pellets (50 µg) were resuspended in freshly prepared $Na_2CO_3$ (100 mM, pH 11 or pH 12). Samples were incubated on ice for 30 min with occasional mixing and subsequently centrifuged at 100,000 g for 30 min. Mitochondrial subfractionation and carbonate extraction samples were TCA precipitated and analysed by SDS- PAGE and immunoblotting.

For crosslinking experiments, mitochondria (0.5–1 mg) were resuspended in import buffer (20 mM HEPES-KOH pH 7.4, 250 mM sucrose, 80 mM KOAc, 5 mM MgOAc, and 5 mM ATP) and incubated with the amino-group-specific homobifunctional and cleavable crosslinker, DSP; dithiobis (succinimidylpropionate) (ThermoFisher) at a final concentration of 0.2 mM for 1 hr at 4°C. Crosslinking reactions were quenched using 100 mM Tris-Cl, pH 7.4 for 30 min at 4°C. Mitochondria were reisolated and solubilised in lysis buffer (20 mM Tris-Cl, pH 7.4, 1 mM EDTA, 1% (w/v) SDS) and boiled at 95°C for 5 min. Clarified samples were diluted with 1% Triton X-100-containing buffer

(1% (v/v) Triton X-100, 20 mM Tris-Cl pH 7.4, 150 mM NaCl, 1X complete protease inhibitor) and incubated with pre-equilibrated anti-FLAG resin for 1 hr at 4°C. Following washing specifically bound proteins were eluted in 2X Laemmli buffer.

## In vitro protein import into isolated mitochondria

Open reading frames (ORF) encoding mitochondrial precursor proteins were cloned into pGEM4Z (Promega). The desired ORFs were amplified by PCR using vector specific primers (M13_FORWARD and M13_REVERSE) and applied to in vitro transcription reactions using the mMessage SP6 transcription kit (Ambion). mRNA was isolated by LiCl precipitation according to the manufacturer's instructions and applied to in vitro translation reactions using rabbit reticulocyte lysate (Promega) in the presence of [$^{35}$S]-methionine/cysteine (Perkin-Elmer). Isolated mitochondria were incubated with translation products in import buffer (20 mM HEPES-KOH pH 7.4, 250 mM sucrose, 80 mM KOAc, 5 mM MgOAc, 10 mM sodium succinate, 1 mM DTT and 5 mM ATP) at 37°C for various times as indicated in the figure legends. Samples subjected to protease treatment were incubated on ice for 10 min with 50 µg/ml proteinase K (Sigma-Aldrich), followed by the addition of 1 mM PMSF and incubation for 10 min on ice. For analysis of protein complexes by blue native electrophoresis and antibody-shift mitochondria were solubilised in 1% digitonin (at 1 mg/mL) and incubated with the desired antibody for 30 min on ice, prior to clarification and subsequent analysis by BN-PAGE. Following in vitro import samples were separated by SDS-PAGE and BN-PAGE and radiolabelled proteins were detected by digital autoradiography.

## Quantification of autoradiographs and statistical analysis

Protein import assays were performed in triplicate unless indicated otherwise. [$^{35}$S]-proteins were visualised using Typhoon PhosphorImage (GE healthcare) and were analysed using ImageJ software (RRID:SCR_003070). The intensity of relevant bands was determined by subtracting the background signal to give the true relative amount of protein (assembled complexes for BN-PAGE analysis and imported proteins for SDS-PAGE analysis). Each specific time point was normalised against the value in control mitochondria with the longest import time point and were computed as a % of this control. Differences in the import of proteins between control (scrambled siRNA) and Tim29 or hTim22 knock-down mitochondria were examined using a paired two-tailed Student's t-test on 2 d.f. based on the three log ratios (control/knockdown) of imported or assembled protein.

## Measurement of mitochondrial oxygen consumption, cell proliferation and cell viability

Mitochondrial oxygen consumption was measured by high-resolution respirometry with an Oxygraph-2K oxygen electrode (Oroboros Instruments, Innsbruck, Austria) as previously described (*Lim et al., 2015*). Intact cells were incubated in 2 mL of Dulbecco's Modified Eagle Medium (DMEM, high glucose, pyruvate, Glutamax) (ThermoFisher) supplemented with 5% v/v fetal bovine serum and 1x penicillin/streptomycin at 37°C. Basal oxygen consumption rates were recorded, followed by the addition of 10 µM carbonyl cyanide p-trifluoromethoxyphenylhydrazone (FCCP) to obtain maximal oxygen consumption rates. Non-mitochondrial oxygen consumption rates were determined by the addition of 5 µM antimycin A and subtracted from basal and maximal rates. Oxygen flux was calculated using DatLab software (version 4.3.4.51, Oroboros Instruments) and expressed as pmol/s/mg of total cell protein. Each assay was performed in triplicate, with significant differences determined using a two-tailed Student's t-test.

Proliferation of control (scrambled siRNA), hTim22 and Tim29 siRNA transfected cells was examined using the bromodeoxyuridine (BrdU) Cell Proliferation ELISA kit (Abcam-ab126556). BrdU reagent was added to cells 48 hr post siRNA transfection and incubated with cells for 24 hr. BrdU incorporation into newly synthesised DNA was measured by absorbance at 450 nm and 550 nm (background) using a FLUOstar OPTIMA *microplate reader* (BMG LABTECH). Assays were performed in triplicate. Cell viability was determined using propidium iodide staining. Briefly, cells were harvested and washed and resuspended in FACS buffer (0.1% BSA diluted in PBS, 2 mM EDTA) followed by incubation with 3.1 µg/mL of propidium iodide (Sigma). Cells were screened by flow cytometry using Flowjo software (RRID: SCR_008520). Plots of the area of forward-scattered light (FSC-A) versus side-scattered light (SSC-A) were used as the cell gating strategy for identification of

any changes in the scatter properties of the cells. Single cell events were identified using the height and width parameters in FSC and SSC. A dot plot using the area of propidium iodide (PI-A) was used to determine the relative proportion of viable and dead cells (%).

## Electrophoresis and immunoblot analysis

For Blue-native gel electrophoresis mitochondrial pellets were solubilised in ice-cold digitonin-containing buffer (1% [w/v] digitonin, 20 mM Tris, pH 7.0, 0.1 mM EDTA, 50 mM NaCl, 10% [w/v] glycerol) at 1 mg/mL and incubated on ice for 15–20 min. Samples were clarified by centrifugation at 16,000 $g$ for 10 min at 4°C, upon which sample buffer (1/10th the volume- 5% [w/v] Coomassie brilliant blue G-250, 100 mM Bis-Tris, pH 7.0, 500 mM ε-amino-n-caproic acid) was added. Samples were separated on 4–16% polyacrylamide gradient gels at 4°C. Thyroglobulin (669 kDa), Ferritin (440 kDa) and Bovine Serum Albumin (132 and 66 kDa) were used as molecular weight markers. SDS-PAGE was performed using gradient Tris-tricine gels as previously described (*Schägger and von Jagow, 1987*). For western blot analysis, gels were transferred to PVDF membranes. Primary antibodies included: anti-FLAG (Sigma; F1804; RRID:AB_262044), cytochrome $c$ (BD Biosciences; 556433; RRID:AB_396417), anti-OPA1 (BD Biosciences; 612606; RRID:AB_399888), hTim23 (BD Biosciences; 611223; RRID:AB_398755), SDHA (Abcam; ab14715; RRID:AB_301433); ANT3 (Abcam; ab154007; RRID:AB_2619664), glutamate carrier (Abcam; ab137614; RRID:AB_2619663), hTim22 (Sigma; T8954; RRID:AB_1858017), Tim29 (Sigma; HPA041858; RRID:AB_10963429); Tim9 (Abcam; ab57089; RRID:AB_945837); NDUFV1 (Proteintech; 11238-1-AP, RRID: AB_2149040); NDUFV2 (Proteintech; 15301-1-AP; RRID:AB_2149048); Tom70 (ProteinTech 14528-1-AP; RRID:AB_2303727); COXIV (Cell Signaling, 4850; RRID:AB_2085424). Antibodies against human Bak, human Tom22, human Tom40, Mfn2, NDUFAF2 and NDUFA9 were kindly provided by Prof. Michael Ryan (Monash University, Australia).

## Immunoprecipitation and mass spectrometry

Standard immunoprecipitation experiments were performed with 0.5–1 mg mitochondria and analysed by SDS-PAGE and immunoblot analysis. For mass spectrometry (MS) experiments, 5 mg of mitochondria were used. Briefly, isolated mitochondria from control cells (empty vector) and cells expressing FLAG tagged proteins were solubilised in digitonin-containing buffer (described above) supplemented with 1X complete protease (Roche) at 2 mg/mL end over end on a rotary wheel for 30–60 min at 4°C. The mitochondrial lysate was cleared by centrifugation at 16,000 $g$ at 4°C for 30 min and diluted in solubilisation buffer (20 mM Tris, pH 7.0, 0.1 mM EDTA, 50 mM NaCl, 10% [w/v] glycerol, 1X complete protease inhibitor) so that a final digitonin concentration of 0.1% was achieved. The cleared lysates were applied to pre-equilibrated anti-FLAG agarose (Sigma-Aldrich) and incubated at 4°C for 60 min under mild agitation. Unbound material was collected and the anti-FLAG resin and bound proteins were washed X 3 in digitonin-containing buffer (0.1% digitonin [w/v], 20 mM Tris, pH 7.0, 0.1 mM EDTA, 50 mM NaCl, 10% [w/v] glycerol, 1X complete protease inhibitor). Bound proteins were eluted with 0.2 M glycine, pH 2.5. Glycine eluted fractions were electrophoretically separated using SDS-PAGE and proteins were visualised by staining with Coomassie Brilliant Blue stain. Gel lanes were cut into 10 x 2 mm bands using a scalpel blade and proteins were reduced, alkylated and trypsinised as described previously (*Keerthikumar et al., 2015*). Briefly, the gel bands were subjected to reduction by 10 mM DTT (Bio-Rad), alkylation by 25 mM iodoacetamide (Sigma), tryptic digestion overnight with 150 ng of trypsin (Promega). Subsequently, the tryptic peptides were further extracted using acetonitrile (50% w/v) and 0.1% trifluoroacetic acid (0.1%). Extracted tryptic peptides were concentrated to ~10 µL by centrifugal lyophilisation and analysed by LC-MS/MS using LTQ Orbitrap Velos mass spectrometer (Thermo Scientific) fitted with nanoflow reversed-phase-HPLC (Model 1200, Agilent). MGF files were generated using MSConvert using peak picking. The MGF files were searched against the NCBI RefSeq database (RRID:AB_2085424) in a target decoy fashion using MASCOT (RRID:AB_2085424 v2.4, Matrix Science, UK). Search parameters used were: fixed modification (carbamidomethylation of cysteine; +57 Da), variable modifications (oxidation of methionine; +16 Da), three missed tryptic cleavages, 20 ppm peptide mass tolerance and 0.6 Da fragment ion mass tolerance. Peptide identifications with mascot ion score greater than the identity score were deemed significant. The relative protein abundance between

the samples was obtained by estimating the ratio of normalised spectral counts (RSc) as previously described (*Keerthikumar et al., 2015*).

## Acknowledgements

We thank Prof. Mike Ryan (Monash University) for generously providing us with antibodies and critical reading of the manuscript, and Prof. Paul Gleeson (The University of Melbourne) for discussion and advice. We are extremely grateful to Prof. Terry Speed (Walter and Eliza Hall Institute for Medical Research) for advice on statistical analysis of data. YK is supported by Melbourne International Fee Remission Scholarship (MIFRS) and Melbourne International Research Scholarship (MIRS). DS is supported by a Biochemistry Fund Fellowship through the Department of Biochemistry and Molecular Biology, The University of Melbourne.

## Additional information

### Funding

| Funder | Grant reference number | Author |
|---|---|---|
| University of Melbourne | Melbourne International Fee Remission Scholarship | Yilin Kang |
| University of Melbourne | Melbourne International Research Scholarship | Yilin Kang |
| University of Melbourne | Biochemistry Fund Fellowship, Department of Biochemistry and Molecular Biology | Diana Stojanovski |

The funders had no role in study design, data collection and interpretation, or the decision to submit the work for publication

### Author contributions

YK, MJB, ML, JL, MM, IA, SK, SM, DS, Conception and design, Acquisition of data, Analysis and interpretation of data, Drafting or revising the article; C-SA, Conception and design, Acquisition of data, Analysis and interpretation of data

### Author ORCIDs

Diana Stojanovski, http://orcid.org/0000-0002-0199-3222

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
