## [Decision Letter]

Thank you for submitting your article "Tim29 is a novel subunit of the human TIM22 translocase and is involved in complex assembly and stability" for consideration by *eLife*. Your article has been reviewed by three peer reviewers, one of whom is a member of our Board of Reviewing Editors, and the evaluation has been overseen Randy Schekman as the Senior Editor. One of the three reviewers has agreed to reveal his identity: Martin van der Laan (Reviewer #2).

The reviewers have discussed the reviews with one another and the Reviewing Editor has drafted this decision to help you prepare a revised submission.

Summary:

Eukaryotes are postulated to have two specialized hetero-oligomeric protein complexes in the mitochondrial inner membrane that mediate protein import: the TIM23 complex is responsible for the import of matrix proteins and some inner membrane proteins and the TIM22 complex inserts proteins with multiple membrane-spanning regions, in particular the large number of mitochondrial metabolite carriers. Mitochondrial metabolism critically depends on these solute carrier proteins in the inner mitochondrial membrane. Global metabolic adaptations of mitochondria are cornerstones of cellular development, stress responses and pathophysiological processes, e.g. in cancer cells. In view of that, surprisingly little is known about the TIM22 protein sorting machinery (carrier translocase) that integrates metabolite carriers into the inner mitochondrial membrane. Therefore, the identification and characterization of a novel subunit of the human TIM22 complex presented in this work by Stojanovski and co-workers is of great importance for our understanding of mitochondrial biogenesis and physiology.

The authors initially identified this novel component, termed Tim29, by a proteomic analysis of affinity purified TIM22 complexes and demonstrate that Tim29 is indeed a genuine subunit of the human carrier translocase crucial for the stability and functionality of this intricate protein sorting machine. Moreover, they provide evidence that Tim29 directly contacts the general preprotein translocase of the outer membrane, the TOM complex, and may thus be involved in the formation of membrane contact sites. This finding represents a major conceptual advance in the field of mitochondrial biogenesis, because it challenges the common dogma that carrier proteins are transported across the mitochondrial intermembrane space via soluble intermediates.

Overall, the manuscript is clearly written, the experiments are of high technical quality and the conclusions are supported well by the data.

Essential revisions:

1) The results of the MS analysis presented in Table 1 are not very informative. What does "number of peptides" exactly mean? I assume it is not unique peptides? Are all the proteins listed that were detected in the analysis or were there more? Are the proteins listed according to their abundance? Thus, the complete set of data of the quantitative mass spectrometry analysis should be available and added to the supplement.

2) Does the knock-down of Tim29 in HEK293T cells have any consequences on cell proliferation and/or survival? Do the authors observe any effects on mitochondrial (energy) metabolism?

3) In Figure 4, the analysis of mitochondrial protein steady state levels reveals the presence of an additional (faster migrating) band stained with antibodies against hTim22. Is this a degradation product of hTim22? If yes, it would suggest that Tim22 is proteolytically degraded if not in complex with Tim29. This may also explain, why no smaller (faster migrating) hTim22 complexes are detected in Tim29-deficient mitochondria (Figure 4)

4) In Figure 4, it can be seen that the knock-down of Tim29 also decreases the levels of the TIM23 complex that mediates the import of presequence-carrying proteins into mitochondria. Does this lead to a reduction of the steady state levels or the import efficiency of TIM23 complex substrates as well?

5) The authors conclude that the critical targeting information of Tim29 resides within the amino-acid residues 17-89, but no protein construct containing only these amino acid residues was analyzed. Is Tim29(aa17-89) targeted to mitochondria?

6) In Figure 7, the authors addressed a coupling function of Tim29 between the hTIM22 complex and the hTOM complex by pull down experiments. Further control proteins such as abundant outer membrane proteins (e.g. VDAC) and inner membrane proteins should be added.

The reverse IP of tagged Tom22 presented in Figure 7 is not convincing. It is not possible to judge whether any of the TIM22 complex subunits gets more enriched than the control proteins SDHA and OPA1.

7) The authors performed chemical crosslinking studies to demonstrate a close proximity of Tim29 to Tom40 of the outer membrane TOM complex. In these experiments, did they also observe crosslinks of Tim29 to other TIM22 complex components that may shed light on the architecture of the human carrier translocase?

8) A quantification of several key results should be provided. Examples are:

The functional data shown in Figure 4AB: It is difficult to judge the extent of the downregulation by eye.

BN-PAGE analyses shown in Figure 5AB: The authors claim in the text that Tim22 ablation affects ANT1 assembly. I fail to see that. If there is a difference, it is very small. Thus, if the authors want to make this claim, they should provide loading controls, quantify the data and provide some statistical measure of the experimental variation.

The same also applies for Figure 6ABC. If I understood the authors correctly they want to say that import of Tim22 is not affected in the Tim29 knock down but assembly of Tim22 into the TIM complex is. This is probably correct, but the claim needs to be supported by a quantitative analysis of the SDS and the BN Gels.

---

## [Author Response]

*Essential revisions:*

*1) The results of the MS analysis presented in Table 1 are not very informative. What does "number of peptides" exactly mean? I assume it is not unique peptides? Are all the proteins listed that were detected in the analysis or were there more? Are the proteins listed according to their abundance? Thus, the complete set of data of the quantitative mass spectrometry analysis should be available and added to the supplement.*

We have removed Table 1 and have now provide the complete data set in Figure 1.

*2) Does the knock-down of Tim29 in HEK293T cells have any consequences on cell proliferation and/or survival? Do the authors observe any effects on mitochondrial (energy) metabolism?*

This point has now been addressed in Figure 4—figure supplement 1. We have looked at the cellular consequences of Tim29 depletion by measuring oxygen consumption in intact cells, cell proliferation and cell survival. We show: [1] Tim29 depletion significantly reduced basal mitochondrial oxygen consumption to 65% as does the KD of hTim22 in HEK293T cells. [2] Using BrdU incorporation as a measure for cell proliferation we have established that the knockdown of both Tim29 and hTim22 has no significant impact on cell proliferation. [3] KD of Tim29 does not influence cell viability, when we measured the uptake of propidium iodide.

*3) In Figure 4, the analysis of mitochondrial protein steady state levels reveals the presence of an additional (faster migrating) band stained with antibodies against hTim22. Is this a degradation product of hTim22? If yes, it would suggest that Tim22 is proteolytically degraded if not in complex with Tim29. This may also explain, why no smaller (faster migrating) hTim22 complexes are detected in Tim29-deficient mitochondria (Figure 4)*

To address this query we monitored the levels of hTim22 over time by western blot in mitochondria isolated from Tim29 KD cells, but established that this lower species was an irregularity of this particular experiment and not a degradation product of hTim22. To avoid confusion we have replaced the entire panel for Figure 4.

*4) In Figure 4, it can be seen that the knock-down of Tim29 also decreases the levels of the TIM23 complex that mediates the import of presequence-carrying proteins into mitochondria. Does this lead to a reduction of the steady state levels or the import efficiency of TIM23 complex substrates as well?*

We have now included Figure 4—figure supplement 2 that shows that the TIM23 substrates (COXIV, NDUFV1 and NDUFV3) are moderately reduced at the steady state level and that the import kinetics of NDUFV1 and NDUFV3 is also reduced. As suggested in point 4 we believe these effects are secondary due to the reduced levels of the TIM23 complex in the Tim29 KD cells.

*5) The authors conclude that the critical targeting information of Tim29 resides within the amino-acid residues 17-89, but no protein construct containing only these amino acid residues was analyzed. Is Tim29(aa17-89) targeted to mitochondria?*

We have generated a construct encoding Tim29aa17-89 fused to a 3XFLAG tag. When expressed in HeLa cells Tim29aa17-89 is targeted to mitochondria. This data has been included in Figure 2. However, as discussed below (Minor Point #1) this construct and C19ord52Δ16 are NOT properly sorted to the inner membrane. Thus, we conclude the first 16 amino acids of C19orf52 (Tim29) are essential for proper mitochondrial sorting.

*6) In Figure 7, the authors addressed a coupling function of Tim29 between the hTIM22 complex and the hTOM complex by pull down experiments. Further control proteins such as abundant outer membrane proteins (e.g. VDAC) and inner membrane proteins should be added.*

*The reverse IP of tagged Tom22 presented in Figure 7 is not convincing. It is not possible to judge whether any of the TIM22 complex subunits gets more enriched than the control proteins SDHA and OPA1.*

We thank the reviewer for raising this point. To address this concern we repeated this experiment using a larger quantity of mitochondria (1.5 mg as opposed to 0.6 mg), to enhance the quality and clarity of data. We have also incorporated a large panel of control antibodies from all four mitochondrial subcompartments to show that TIM22 subunits are coming down specifically. This is now shown in Figure 8.

*7) The authors performed chemical crosslinking studies to demonstrate a close proximity of Tim29 to Tom40 of the outer membrane TOM complex. In these experiments, did they also observe crosslinks of Tim29 to other TIM22 complex components that may shed light on the architecture of the human carrier translocase?*

Following the suggestion of the reviewer we now show crosslinks between Tim29 and Tim9 (displayed in Figure 9) and crosslinks between Tim10b and Tim29 and Tim10b with Tim9 (displayed in Figure 9). Although we did not detect a crosslink between hTim22 and Tim29 we believe this is due to the lack of primary amines in the predicted Tim29 transmembrane domain, which are essential for the amine-reactive crosslinker DSP (used in our experiments). This new data has significantly extended our understanding of the architecture of the humanTIM22 complex.

*8) A quantification of several key results should be provided. Examples are:*

*The functional data shown in Figure 4AB: It is difficult to judge the extent of the downregulation by eye.*

Quantifications have now been included and displayed graphically below the panels of Figure 4.

*BN-PAGE analyses shown in Figure 5AB: The authors claim in the text that Tim22 ablation affects ANT1 assembly. I fail to see that. If there is a difference, it is very small. Thus, if the authors want to make this claim, they should provide loading controls, quantify the data and provide some statistical measure of the experimental variation.*

As suggested we have now included loading controls and quantifications for these gels (displayed beneath BN-PAGE gels in Figure 5), which do indeed support the finding that ANT1 assembly is reduced in the absence of hTim22, but not Tim29.

The same also applies for Figure 6ABC. If I understood the authors correctly they want to say that import of Tim22 is not affected in the Tim29 knock down but assembly of Tim22 into the TIM complex is. This is probably correct, but the claim needs to be supported by a quantitative analysis of the SDS and the BN Gels.

We have now included loading controls for the BN-PAGE experiments and quantitative analysis of the BN-PAGE (displayed beneath BN-PAGE gels in Figure 6) and SDS-PAGE gels (now Figure 7 (NOT 6C)). Given the importance of the observation that hTim22 assembly is significantly reduced in the Tim29 KD mitochondria we analysed these experiments using a two-tailed Student’s t-test (outlined within the text) and obtained a t[2]= 6.02 (p < 0.05), demonstrating a significant reduction in the assembly of hTim22 into the mature complex in Tim29 KD mitochondria. To the contrary, mitochondria isolated from hTim22 KD consistently displayed a moderate increase (~20%) in assembled TIM22 complex in (t[2]= -5.07, p < 0.05).